# The Safety Assessment of Mutagenicity, Acute and Chronic Toxicity of the *Litsea martabanica* (Kurz) Hook.f. Water Leaf Extract

**DOI:** 10.3390/toxics12070470

**Published:** 2024-06-28

**Authors:** Weerakit Taychaworaditsakul, Suphunwadee Sawong, Supaporn Intatham, Sunee Chansakaow, Teera Chewonarin, Phraepakaporn Kunnaja, Kanjana Jaijoy, Absorn Wittayapraparat, Pedcharada Yusuk, Wannaree Charoensup, Seewaboon Sireeratawong

**Affiliations:** 1Clinical Research Center for Food and Herbal Product Trials and Development (CR-FAH), Faculty of Medicine, Chiang Mai University, Chiang Mai 50200, Thailand; weerakit.tay@cmu.ac.th (W.T.); suphunwadee.sa@cmu.ac.th (S.S.); intatham_s@outlook.com (S.I.); 2Department of Pharmacology, Faculty of Medicine, Chiang Mai University, Chiang Mai 50200, Thailand; 3Department of Biochemistry, Faculty of Medicine, Chiang Mai University, Chiang Mai 50200, Thailand; teera.c@cmu.ac.th; 4Department of Pharmaceutical Sciences, Faculty of Pharmacy, Chiang Mai University, Chiang Mai 50200, Thailand; sunee.c@cmu.ac.th (S.C.); wannaree.charoensup@cmu.ac.th (W.C.); 5Department of Medical Technology, Faculty of Associated Medical Sciences, Chiang Mai University, Chiang Mai 50200, Thailand; phraepakaporn.k@cmu.ac.th; 6McCormick Faculty of Nursing, Payap University, Chiang Mai 50000, Thailand; joi.kanjana@gmail.com; 7Highland Research and Development Institute (Public Organization), Chiang Mai 50200, Thailand; absornw@hrdi.or.th (A.W.); npedcharada@gmail.com (P.Y.)

**Keywords:** *Litsea martabanica* leaf extract, medicinal plants, acute toxicity, chronic toxicity, safety evaluations, environment and human health

## Abstract

*Litsea martabanica* (Kurz) Hook.f. has traditionally been used as an anti-insecticidal agent and as a medication due to its hepatoprotective properties by highland communities in Thailand. This study examined the mutagenicity, as well as the acute and chronic toxicity, of the *L. martabanica* water leaf extract in Sprague-Dawley rats. The pharmacognostic evaluation of *L. martabanica* was performed in this study to ensure its authenticity and purity. Then, the sample was extracted using decoction with water to obtain the crude water extract. The assessment of acute toxicity involved a single oral administration of 5000 mg/kg, whereas the chronic toxicity assessment comprised daily oral doses of 250, 750, and 2250 mg/kg over 270 days. Various physiological and behavioral parameters, as well as body and organ weights, were systematically monitored. The endpoint assessments involved hematological and biochemical analyses plus gross and histopathological assessments of the internal organs. Our results exhibited no mutagenic activation by the *L. martabanica* water leaf extract in the Ames test, and no acute toxicity was observed. In the chronic toxicity tests, no abnormalities were found in rats receiving the *L. martabanica* water leaf extract across multiple measures, comprising behavioral, physiological, and hematological indices. Crucially, the histopathological assessment corroborated previous studies, reporting an absence of any tissue abnormalities. The results revealed that the *L. martabanica* water leaf extract had no adverse effects on rats over 270 days of oral administration. This demonstrates its safety and crucial scientific evidence for informing public policy and enabling its potential future commercial use in both highland and lowland communities.

## 1. Introduction

*Litsea* belongs to the Lauraceae family, a group of polymorphic arboreal or shrub species encompassing over 400 varieties and predominantly distributed across tropical Asia, the Pacific islands, Australia, and Central and North America, of which 27 species have been identified in Thailand. Revered in traditional medical practices for several centuries, 20 distinct *Litsea* species within the Chinese pharmacopeia have been acknowledged for their efficacy in addressing a spectrum of maladies, ranging from gastrointestinal disturbances to rheumatic conditions [1]. One species, *L. cubeba*, a taxonomically defined member of this genus, harnesses various botanical components, including bark and leaves, in the therapeutic modulation of multifarious ailments. This botanical entity has substantiated its pharmacological significance through the manifestation of noteworthy attributes, such as antioxidant [2], anticancer [3], anti-inflammatory [4], anti-insecticidal [5], and hepatoprotective properties [1]. 

In previous studies, we examined the efficacy of both the root and the leaf of *Litsea martabanica* (Kurz) Hook.f. in terms of its insecticidal and hepatoprotective properties [6,7]. We also focused on the environmental damage resulting from harvesting *L. martabanica*, as cutting down the entire tree to use some parts for medicinal purposes, which has been the standard practice in some communities, can cause environmental damage and reduce the species’ future availability. Both the root and the leaf are used by highland communities in traditional medications [6,7]. Although *L. martabanica* has demonstrated health benefits, there are concerns about its safety with regular use. There is limited evidence regarding the short- and long-term effects of *L. martabanica* use. In a previous study, a preliminary evaluation of different concentrations of the extract was conducted using the Hippocratic screening method [8]. In the study, two female rodents were given a single administration of the *L. martabanica* leaf extract at a concentration of 5000 mg/kg. The findings after 24 h indicated that the administered dosage did not result in toxicity or mortality among the animals used in the experiment. 

Toxicology tests play a pivotal role in the overall advancement of pharmacological research and drug development. Before clinical use in humans, establishing a drug’s safety is a primary objective of animal toxicology testing [9,10,11]. Consistent with the 2019 Global Report on Traditional and Complementary Medicine by the World Health Organization (WHO), herbal products commonly undergo thorough safety assessments resembling those required for standard pharmaceuticals, including post-market monitoring [12]. In addition to toxicity tests, mutagenicity testing is also an essential assessment for herbal products that is used to examine their efficacy and safety in humans and animals, as well as their impact on the environment. The test results can significantly impact the product registration, influencing approval by regulatory agencies such as the Food and Drug Administration (FDA) and the Environmental Protection Agency (EPA) [13]. Government regulatory agencies typically require animal toxicity studies to be conducted in accordance with the Organization for Economic Co-operation and Development (OECD) for the Principles of Good Laboratory Practice (GLP) before granting herbal medicine registration [12]. Before investigating both the efficacies and toxicities, a pharmacognostic evaluation of the crude drug is conducted to verify its authenticity and purity and to establish a scientific database for future use. Monitoring the quality of raw materials is crucial to ensure high-quality future extracts. Acute toxicity tests are typically performed during the initial 24 hours following the administration of a high dose of a drug to detect any adverse effects that may manifest in the test animals. Although some drugs do not show significant adverse effects during acute toxicity tests, they can still potentially result in sub-chronic or chronic toxicity [10,14,15,16]. For that reason, chronic toxicity tests are usually also required to include assessments of the physiological, hematological, biochemical, and pathological safety of drug use [17]. 

The aim of this study was to examine both the acute and chronic toxicities of the *L. martabanica* water leaf extract in Sprague-Dawley rats. This research is intended as a preliminary step in the clinical testing of the *L. martabanica* leaf in humans and its potential for integration into basic community healthcare systems. Additionally, the study aimed to help expand accessible healthcare for the general population by vetting the traditional healthcare practices, which are part of the wisdom of highland communities, through evaluations using scientific processes. The demonstration of its safety through mutagenicity, as well as through acute and chronic toxicity studies, provides important scientific evidence that can both inform public policy and pave the way for commercial use among both highland and lowland populations. 

## 2. Materials and Methods

### 2.1. Chemicals and Reagents

Dimethyl sulfoxide (DMSO), D-glucose-6-phosphate disodium salt, nicotinamide adenine dinucleotide phosphate sodium salt (NADP), magnesium chloride, L-histidine monohydrate, D-biotin, sodium azide (NaN_3_), 2-aminoanthracene (2-AA), 2-aminofluorene (2-AF), 2-amino-3-methyl-3H-imidazo[4,5-f] quinoline (IQ), and 2-amino-l-methyl-6-phenylimidazo[4,5-b] pyridine (PhIP) were purchased from Sigma Chemical Co (St. Louis, MO, USA). Anhydrous dibasic potassium phosphate, D-Glucose, magnesium sulfate, citric acid monohydrate, monobasic sodium phosphate, sodium ammonium phosphate, dibasic sodium phosphate, and sodium chloride were purchased from Merck (Whitehouse Station, NJ, USA).

### 2.2. Plant Material

The leaves of *Litsea martabanica* were harvested from the Chiang Mai province, Thailand. The identification of the plant material was performed by the taxonomist and compared with the voucher specimen deposited in the Queen Sirikit Botanical Garden (No. WP 7185). The samples were cut into pieces, dried in the hot air oven until the moisture was less than 10%, and then they were pulverized. The quality of the raw material was assessed according to the methods outlined in the 2018 Thai Herbal Pharmacopoeia [18].

### 2.3. Quality Control of Raw Material

The leaf powder of *L. martabanica* was assessed for its physico-chemical properties following the methods expressed in Thai Herbal Pharmacopoeia 2018 (THP2018), including the determination of weight loss on drying, total ash, and extractive values [18]. Detailed descriptions of the methods are described in a previous study [19].

#### 2.3.1. Determination of Loss on Drying

The powdered drug samples, ranging from 2 to 5 g, were precisely weighed and then put into weighing bottles. After that, they were dried at 105 °C in a hot air oven until their weight remained constant. Then, the samples were allowed to cool in a desiccator. The moisture content was then calculated by measuring the weight loss.

#### 2.3.2. Determination of Ash Values

##### Determination of Total Ash

Approximately 2–3 g of the powdered drug was meticulously weighed and added to a pre-dried and pre-weighed crucible. The test sample underwent gradual heating in an advanced electrical muffle furnace (Thermo Fisher, Waltham, MA, USA) until it reached 500 °C, ensuring the complete removal of carbon. Following this, the sample was cooled in a desiccator and reweighed to precisely measure the total ash content.

##### Determination of Acid-Insoluble Ash

The ash was treated with 25 milliliters of 2 M hydrochloric acid (HCl) in a crucible. The crucible, covered with a watch glass, was then heated in a water bath for 5 minutes. The insoluble material was filtered using No. 41 filter paper, and the filtrate was washed with hot water until it reached a neutral pH. The filter paper with the insoluble material was then placed in a crucible and heated gradually in an electrical muffle furnace until the weight became constant. Finally, the test samples were weighed after cooling in desiccators to determine the amount of acid-insoluble material.

#### 2.3.3. Determination of Extractive Value

##### Ethanol Soluble Extractive Value

A total of 5.0 g of the powdered drug was transferred to a glass-stoppered conical flask. The powder was then macerated with 100.0 mL of 95% ethanol for 6 h, shaken frequently, and allowed to stand for 18 h. Afterward, the sample was filtered, and the residual material was washed with ethanol to make the filtrate up to 100.0 mL. Subsequently, 20.0 mL of the filtrate was transferred to a pre-weighed evaporating dish. The test sample was then evaporated to dryness in a water bath, dried at 105 °C, cooled in a desiccator, and then weighed.

##### Water-Soluble Extractive Value

The procedure involved mixing approximately 5.0 g of powdered drug with 100 mL of water and shaking the mixture for 6 h. The mixture was then left to stand for 18 h. Afterward, the sample was filtered, and the solid residue was washed with water to produce 100 mL of filtrate. From the filtrate, 20 mL was transferred to a pre-weighed evaporating dish. The sample was then evaporated to dryness in a water bath, dried at 105 °C, cooled in a desiccator, and then weighed.

### 2.4. Extract of L. martabanica (Leaf)

The leaves of *L. martabanica* were extracted using traditional methods, specifically through decoction with water as the solvent. The powder from the *L. martabanica* leaf was boiled in water for 1.5 h using a ratio of 1 L of water to 100 g of leaves that had been presoaked in water for 30 min. The filtrate was separated from the marc by filtering through the filter paper, and the remaining plant material was boiled two more times for 30 min each time to ensure the complete extraction of all the components. The filtrate was then collected and concentrated to approximately 3% Brix, and the acid–base balance was measured. The extract was then dried using a spray dryer (BUCHI Mini Spray Dryer B-290, BUCHI Labortechnik AG, Flawil, Switzerland) at an inlet temperature of 140 °C and an outflow temperature of 85 ± 5 °C, with a 100% aspirator and 25% pump, to obtain a water-soluble powder.

### 2.5. Mutagenicity and Antimutagenicity Assessment

Mutagenicity and antimutagenicity assessments of the *L. martabanica* water leaf extract were conducted in duplicate using the preincubation method, as previously described [20,21], both with and without the S9 mix. *Salmonella typhimurium* strains TA98 and TA100 purchased from the American Type Culture Collection (ATCC, Manassas, VA, USA) were used as the tester strains.

For the mutagenicity test, the *L. martabanica* water leaf extract was dissolved in sterilized normal saline to concentrations ranging from 125 to 1000 µg. In all the subsequent tests, the highest non-toxic dose or the lowest toxic dose identified in this initial assay determined the upper limit of the dose range examined. The toxicity was identified through a decrease in the number of histidine revertants (His^+^). These concentrations were added to either 0.5 mL of 0.2 M phosphate buffer or 0.5 mL of 4% S9 mixture, along with 0.1 mL of bacterial culture, and were incubated at 37 °C for 20–30 min. Following the incubation, 2 mL of top agar was added, and the mixture was poured onto plates containing minimal agar. The plates were then incubated at 37 °C for 48 h and the His + revertant colonies were manually counted. All the experiments were conducted in triplicate. 2-aminofluorene (2-AF) was used as the standard mutagen in the experiments without the S9 mix for TA98 and TA100. In the experiments with S9 activation, 2-aminoanthracene (2-AA) was used with TA98 and TA100. DMSO (100 μL/plate) served as the negative (solvent) control.

For the antimutagenicity, four different concentrations of extract were combined with recognized mutagens in assays conducted with and without metabolic activation, utilizing the *S. typhimurium* tester strains TA98 and TA100. In the assays without metabolic activation, 2-AF was employed for TA98 and TA100, while PhIP was used for TA98, and IQ was used for TA100 tested in the presence of metabolic stimulation. The extract was blended with either 0.5 mL of 0.2 M phosphate buffer or 0.5 mL of 4% S9 mixture for the metabolic activation, along with 0.1 mL of bacterial culture and the mutagen, and then incubated at 37 °C for 20–30 min. Following the incubation, 2 mL of top agar was introduced, and the content of each tube was gently mixed and poured onto a glucose minimal agar plate. After the top agar solidified, the plates were incubated for 48 h at 37 °C, and the count of the revertant colonies per plate was conducted.

### 2.6. The Animal Subjects and Ethical Considerations

In this investigation, male and female Sprague-Dawley rats weighing between 180 and 200 g were used. The animals were purchased from Nomura Siam International (Nomura Siam International Co., Ltd., Bangkok, Thailand). The environment was kept at a temperature of 25 ± 1 °C with 60% relative humidity, and a 12-h cycle of light and darkness. The rats had unrestricted access to water and food. For the animal welfare insurance, each rat was given at least one week to acclimate before the experiment began. This study adhered to ethical guidelines and was approved by the Research Ethics Committee for Animal Studies at the Faculty of Medicine, Chiang Mai University, Thailand (approval code: 10/2563).

### 2.7. Hippocratic Assessment

Assembling upon the recognized methodologies from a previous study [8], the present research used the Hippocratic assessment to investigate the safety information of the *L. martabanica* water leaf extract. In acute toxicity, five female rats were given a single oral dose of 5000 mg/kg and an individual observation in open fields at intervals of 5, 10, 15, 30, 60, 120, and 240 min post administration, and then once daily for 24 h. In chronic toxicity (10 females and 10 males of each group), were given a single oral dose of 250, 750, and 2250 mg/kg, respectively, for 270 days. The satellite group received the extract at a dose of 2250 mg/kg for 270 days with a 28-day observation. The observations included motor activity (measured using infrared beams in a motor activity cage), respiration rate (counting breaths per minute), righting reflex (noting the animal’s ability to flip over), and screen grip (assessing the animal’s ability to cling to a cage with both their hindlimbs and forelimbs). The purpose of this assessment was to detect any probable adverse reactions, such as vomiting, sedation, watery diarrhea, and muscle spasms.

### 2.8. Acute Toxicity Assessment

Following the WHO and OECD 420 Guideline [22,23], the randomized rats were allocated to either a treatment group (*n* = 5) receiving a single oral dose of 5000 mg/kg *L. martabanica* water leaf extract or a control group (*n* = 5) receiving 2 mL/kg distilled water, both administered via an oral gavage. Acute toxicity signs, including fatigue, vomiting, diarrhea, and muscle spasms, were monitored during the first six hours and thereafter every day for 14 days. Body weight was noted on days 7 and 14, along with all death incidents. After the observation period, the rats were sacrificed using an intraperitoneal injection of thiopental sodium (120 mg/kg). A physical examination, including vital signs, reflexes, and pulse, was conducted for the verification of death, followed by a gross pathology of the internal organs, consisting of the heart, kidneys, lungs, and liver. The collected organs were conserved in 10% formaldehyde for additional examinations.

### 2.9. Chronic Toxicity Assessment

The chronic toxicity assessment followed the WHO and OECD 452 Guideline [17]. The male and female rats were separated into six groups. Each group contained 10 animals of each sex, except for groups 2 and 6, which had five animals of each sex. Group 1 (control) was orally administered distilled water at a dose of 2 mL/kg daily for 270 days. Group 2 (satellite group) received the same treatment as the control group for 270 days, followed by an additional 28-day observation period. Groups 3–5 were treated with daily oral doses of the *L. martabanica* water leaf extract at 250, 750, and 2250 mg/kg, respectively, for 270 days. Group 6 (satellite group) received 2250 mg/kg of the extract for 270 days with a 28-day post-treatment observation period. Behavioral appearances, abnormal clinical signs, and body weight were observed throughout. The dead animals underwent immediate necropsy. On day 270, all the rats were euthanized with 120 mg/kg thiopental sodium using an intraperitoneal injection, after which their vital signs, pulse, and reflexes were checked for confirmation of the death of the animals. Blood samples were collected to analyze their hematological and biochemical characteristics. Both macroscopic and microscopic examinations of the internal organs were performed, including the lungs, heart, liver, pancreas, kidneys, stomach, intestines, adrenal glands, eyes, brain, muscles, and nerves.

### 2.10. Statistical Analysis

The data are presented as mean ± S.E.M. The data analysis of the acute toxicity assessment employed either the *t*-test or the Mann–Whitney U test, as suitable. Whereas in the chronic toxicity assessments, an initial examination was conducted using the Shapiro–Wilk test to evaluate the normality. If the Shapiro–Wilk test revealed no significant deviation from a normal distribution, we proceeded with an ANOVA followed by Tukey’s multiple comparison tests. On the other hand, if the Shapiro–Wilk test indicated a deviation from the normality, we utilized the Kruskal–Wallis nonparametric ANOVA test followed by Dunn’s test. The statistical significance was set at *p* < 0.05. The statistical analyses were carried out using IBM SPSS Statistics, version 22.0 (International Business Machines Corporation, Armonk, NY, USA).

## 3. Results

### 3.1. Specification of Crude Drug of L. martabanica Leaf

#### 3.1.1. Macroscopic Identification

The crude drug of *L. martabanica* includes the whole leaf and the leaf broken into pieces. The petiole is 0.9–1.5 cm long and is densely covered with short, soft, yellow-brown hairs. The leaves are narrowly oval, 3.5–4.5 cm wide, and 11–13.5 cm long, tapering to a point at the tip, and blunt at the base. The ventral side of the leaf is dark green to brownish and quite smooth. There are short, soft, yellow-brown hairs covering the midrib and the base of the leaf veins. The leaf veins are reticulated, feathery, and brownish black. The upper midrib is concave and grooved, and the underside is convex. The leaf veins are in 6–8 pairs. The leaf margins are smooth and quite tough in texture (Figure 1).

#### 3.1.2. Microscopic Identification

The microscopic characteristics of the leaf of *L. martabanica* are presented in Figure 2.

#### 3.1.3. Physical and Chemical Identification

The crude drug’s physiochemical examinations, including weight loss on drying, total ash, acid-insoluble ash, ethanol-soluble extractive value, and water-soluble extractive value, along with their mean values, are presented in Table 1.

The leaf of *L. martabanica* was extracted using decoction with water as a solvent, following the traditional methods. The chemical composition of the *L. martabanica* water leaf extract was thoroughly analyzed using phytochemical screening [24], which revealed the presence of phenolics, flavonoids, saponins, and terpenoids. Although a monograph of *L. martabanica* is not included in any pharmacopeia or textbooks, this study includes a description of the *L. martabanica*’s leaf alignment using the methods outlined in the Thai Herbal Pharmacopeia. Providing such a detailed description can help to ensure quality control when handling raw materials in future studies.

### 3.2. Mutagenicity Assessment

Table 2 shows the results of the mutagenicity assays. Potent mutagenicity was observed in both the 2-AA and 2-AF treated groups, with a significant increase in the number of revertant colonies compared with the DMSO group. The assays found that the *L. martabanica* water leaf extract at concentrations ranging from 125 to 1000 µg/mL showed no toxicity to the bacterial cells and did not exhibit mutagenic effects on the *S. typhimurium* strains TA98 or TA100, either with (+S9) or without (−S9) metabolic activation enzymes.

### 3.3. Antimutagenicity Assessment

The assessment of the antimutagenic effect of the *L. martabanica* water leaf extract at concentrations ranging from 125 to 1000 µg/mL found that, in the presence of metabolic activation, the *L. martabanica* water leaf extract significantly exhibited antimutagenic activity in the *S. typhimurium* strains TA98 and TA100 when stimulated by the mutagenic substances PhIP and IQ. In the absence of metabolic activation, the extract significantly exhibited antimutagenic activity only in the *S. typhimurium* strain TA100 when stimulated by 2-AF; however, the extract did not exhibit any antimutagenic effect in the *S. typhimurium* strain TA98 induced by 2-AF under the same conditions without S9 (Table 3).

### 3.4. Hippcratic Assessment

The Hippocratic assessments were directed to evaluate the possible toxicity of the oral administration of the *L. martabanica* water leaf extract that was administered at a dosage of 5000 mg/kg in female rats. In the chronic toxicity, the extract was administered to the rats at a dosage of 250, 750, and 2250 mg/kg for 270 days, and the satellite group received 2250 mg/kg for an additional 28 days. The observations showed no negative behavioral changes or deaths during that period. Furthermore, a gross investigation of the internal organs and all the carcasses revealed no abnormalities.

### 3.5. Acute Toxicity Assessment

As recommended in the OECD and WHO guidelines, an acute assessment was performed using a single oral dose of 5000 mg/kg of the *L. martabanica* water leaf extract administered to the female rats. No abnormal behavior was observed in the experimental or control groups during the initial 24-h observation period. The characteristics of eating, body weight, and excretion of the rats in the experimental groups were not different from the control group. No mortalities were recorded in any of the groups during the study. Body weight measurements on days 0, 7, and 14 revealed no statistically significant differences from the control group (Table 4).

On day 15, the rats were euthanized, and a comprehensive gross examination of the internal organs found no differences between the treated groups and the control group. The effect of the *L. martabanica* water leaf extract on organ weights is shown in Table 5. The internal organ weights, including the brain, heart, lungs, liver, adrenal gland, kidney, ovary, and uterus, of the rats that received the *L. martabanica* water leaf extract at 5000 mg/kg had no significant differences compared with the control group. A visual necropsy of the internal organs of the rats did not reveal any abnormalities in the color and size of the internal organs among the groups. The data on animal health, symptoms monitoring, and the autopsy results of the rats showed that, in the female rats, the receipt of a single oral dose of 5000 mg/kg body weight of the *L. martabanica* water leaf extract did not cause any acute toxicity.

### 3.6. Chronic Toxicity Assessment

#### 3.6.1. Body Weight

Both the female and male rats were orally administered daily doses of either 250, 750, or 2250 mg/kg bw of the *L. martabanica* water leaf extract for 270 days. No abnormal behavior was observed in any of the treated groups or in the control groups. The characteristics of eating, body weight, and excretion of the rats in the treated groups were not different from those of the control group. No mortalities were noted in any of the groups during the study.

The study evaluated the body weight of the female and male rats treated with the *L. martabanica* water leaf extract, as shown in Table 6 and Table 7, respectively. The results showed that the female rats’ body weight in the satellite control group and in the group receiving 750 mg/kg bw of the *L. martabanica* water leaf extract had significantly decreased on day 30 compared with the control group. The body weight of the male rats that received the *L. martabanica* water leaf extract at a dose of 750 mg/kg bw had increased significantly on day 30 and the body weight of the satellite treatment group of the male rats had significantly increased on days 30, 120, 180, and 240 compared with the control group.

#### 3.6.2. Organs Weight

The relative organ-to-body weight ratios (organ weight/body weight × 100) were evaluated in both the female and male rats, as shown in Table 8 and Table 9, respectively. A reduction in the kidney weight was observed in both the female rats receiving the *L. martabanica* water leaf extract at a dose of 750 mg/kg bw (Table 8) and in the satellite treatment group (Table 9) compared with the control group.

#### 3.6.3. Hematology and Serum Biochemistry Assessment

The hematological assessment of the female and male rats, as shown in Table 10 and Table 11, respectively, revealed no changes in the counts of red blood cells, hemoglobin, hematocrit, mean corpuscular volume, mean corpuscular hemoglobin, mean corpuscular hemoglobin concentration, or platelets.

In the differential white blood cell (WBC) count of the female rats, as shown in Table 10, it is evident that the group administered the *L. martabanica* water leaf extract at a dose of 2250 mg/kg exhibited an increase in the neutrophil count compared with the control group. Conversely, there was a notable reduction in lymphocytes among the rats receiving the *L. martabanica* water leaf extract at doses of 250 and 750 mg/kg compared with the control group. Furthermore, the group receiving 750 mg/kg demonstrated a significant decrease in eosinophil counts. Regarding the WBC count differential of the male rats (Table 11), the groups administered the *L. martabanica* water leaf extract at a dose of 750 mg/kg had significantly decreased WBC counts. The lymphocyte count exhibited a dose-dependent decrease in both the *L. martabanica* water leaf extract-treated groups and the satellite treatment group compared with the control group.

The serum biochemistry assessment of blood in the female rats revealed a significant increase in albumin levels among rats receiving the *L. martabanica* water leaf extract at a dose of 750 mg/kg compared with the control group (Table 12). Furthermore, the total bilirubin levels were significantly elevated in the groups receiving the *L. martabanica* water leaf extract at doses of 250 and 750 mg/kg compared with the control group. The satellite treatment group exhibited a statistically significant increase in aspartate transaminase (AST) levels compared with the control group. In the clinical chemistry values of the male rats, it was observed that the group receiving the *L. martabanica* water leaf extract at a dose of 750 mg/kg experienced reduced creatinine levels (Table 13). Additionally, the group receiving 2250 mg/kg demonstrated a statistically significant decrease in both blood urea nitrogen (BUN) and creatinine levels compared with the control group.

#### 3.6.4. Necropsy and Histopathological Assessment

In the necropsy assessment and the visual observation of the internal organs in both the female and male rats that received the *L. martabanica* water leaf extract, including the lungs, heart, spleen, liver, adrenal gland, kidney, stomach, pancreas, intestines, thymus gland, muscle and nerves, eyes, brain, ovaries, uterus, testis, epididymis, and vertebral bones, no abnormalities were detected. The size, shape, and color of the organs were normal when compared with the control group. The histopathological assessment of the internal organs of both the female and male rats did not reveal any signs of tissue damage caused by the *L. martabanica* water leaf extract (Figure 3 and Figure 4). Specifically, in the heart, the myocardial fibers were orderly arranged without any disruptions. The liver cells appeared structurally normal with no signs of necrosis or inflammatory cell infiltration. The microscopic examination of the kidneys showed clear and intact glomerular vascular collaterals, well-preserved renal tubules, clear lumens, intact basement membranes, neatly arranged epithelial cells, and an absence of vacuolated cell formation or inflammatory cell infiltration. Similarly, the lung tissue exhibited a normal structure, an intact alveolar septum and lumen, and a uniform distribution of the lung interstitium.

## 4. Discussion

In pharmacological and toxicological studies, the standardization of plant extracts is an important part of the research to obtain accurate research data that can be beneficial to humans in the future [25]. Firstly, the quality of the raw material will need to be assessed, including the evaluation of organoleptic, macroscopic, microscopic, physical, and chemical properties. Nevertheless, as shown in Figure 1, the crude drug can be recognized based on the morphology of the plants. The identification process, however, becomes much more problematic when the plant material is in powder form.

The powder of the *L. martabanica* leaf contains various tissues, including unicellular trichomes, the lower epidermis with tetracytic stomata, idioblasts, and the epidermis with cicatrix, as shown in Figure 2, which were used for identification purposes. Quality evaluations of the crude drug were performed using various physicochemical examinations, including total ash, acid-insoluble ash, ethanol-soluble extractive value, water-soluble extractive value, and loss on drying following the established procedures in THP 2018. The specifications of the crude drug of *L. martabanica* reported in this study are appropriate criteria for evaluating the quality of the raw material before further processing.

The plant material was extracted using decoction with water, imitating the traditional method used to obtain the crude water extract. The chemical composition of the *L. martabanica* water leaf extract was assessed using phytochemical screening, which revealed the presence of phenolics, flavonoids, saponins, and terpenoids in both the crude drug and its extract. The extract used in this study had the same bioactive compositions as the previous study [6,7]. A review of the literature found that phenolics, flavonoids, saponins, and terpenoids have been reported to have antitumor activity [26], anti-inflammatory activity [27], antimicrobial activity [28], and cardiovascular protective activity [29]. In particular, terpenoids have been found to have antioxidant effects, enhance acetylcholinesterase (AChE) activity, and improve the anomalies of liver histopathology [7]. These specifications can be used as a reference for future research on extract preparation process repeatability. Although the bioactive compound of this plant could not be identified, it will be studied in the future via bioassay-guided isolation to identify its specific chemical constituent as a chemical marker or biomarker.

The mutagenicity and antimutagenicity assessments using the *Salmonella typhimurium*, commonly known as the Ames test, is a widely recognized short-term bacterial assay used to identify substances that have the potential to induce genetic damage, leading to gene mutations [30,31]. Consistent with the Food and Drug Administration (FDA) in Thailand, mutagenicity tests such as the Ames test can provide safety data and other evidence for consideration before the registration of herbal products [32]. To evaluate the mutagenic properties of the *L. martabanica* water leaf extract, both with and without S9, an enzyme preparation derived from the rat liver was used for metabolizing various toxic substances. Our results showed that the *L. martabanica* water leaf extract was not mutagenic. A previous study of a plant in the same genus found that *L. cubeba* oil was not mutagenic in the *S. typhimurium* strains TA98, TA100, TA1535, or TA1537, either with or without metabolic S9 activation [33]. In addition, the antimutagenicity activity of the *L. martabanica* water leaf extract was also screened against known mutagens using the *Salmonella* TA98 and TA100 strains. The results showed that the *L. martabanica* water leaf extract exhibited antimutagenic activity in the *S. typhimurium* strains TA98 and TA100 when stimulated using the mutagenic substances PhIP and IQ in combination with the S9 fractions. Additionally, in the TA98 and TA100 mutations induced by 2-AF under conditions without S9, no antimutagenic effect was exhibited. Research has shown that the *L. martabanica* water leaf extract at concentrations ranging from 125 to 1000 µg/mL exhibits a moderate anti-mutagenic effect and does not induce genetic toxicity.

The acute oral toxicity assessment plays a crucial role in evaluating the immediate negative effects that may occur within 24 h after administering a significant single oral dose of a substance [22]. In this study, we administered a high dose (5000 mg/kg) of the *L. martabanica* water leaf extract to evaluate its safety. Our findings show that the administration of 5000 mg/kg bw of the *L. martabanica* water leaf extract did not lead to abnormal behavior (Appendix A) or significant changes in body weight compared with the control group. Additionally, a gross examination of the internal organs did not reveal any abnormalities in the shape or color. Furthermore, the extract did not significantly alter the organ weight compared with the control group, and no mortality in any of the groups was observed during the study. Due to the toxicity assessments, if it does not show toxicity in the female test animals, it is unlikely to show toxicity in the male test animals [22]. The experiment revealed that when the extract was administered to the female test animals, no signs of toxicity were observed. Hence, it is reasonable to predict that the same dose of the extract would not cause toxicity in males. Consequently, it was concluded that the extract at a dosage of 5000 mg/kg bw did not induce toxicity, suggesting the absence of acute toxicity.

Chronic toxicity assessments that extend six months to two years can evaluate the long-term effects of prolonged substance exposure and can identify delayed adverse effects. These assessments should align with both the WHO and OECD guidelines, which recommend replicating the expected duration of human clinical exposure in animal models [17,23]. Assessing changes in behavior and body weight is a principle and crucial method for detecting toxicity. For example, A 10% or greater reduction in the initial body weight of test animals may signify significant adverse effects and may indicate potential risks to survival [34,35,36,37]. In this study, long-term exposure (around 270 days) to the *L. martabanica* water leaf extract did not lead to significant signs of toxicity (Appendix A) and changes in body weight in male or female rats. These findings strongly support the long-term safety of using this extract in rat models, positioning the establishment of its use in future clinical research applications.

Changes in internal organ weight can serve as a vital indicator for evaluating the effects of drug experience on organ reliability, particularly in toxicological assessments requiring a systematic comparison between treated and untreated animals [38,39]. In the present study, we observed that relative organ weights showed no abnormal values, with the exception of significant changes in the kidneys. Nevertheless, differences in the experimental animals themselves could be the cause of this peculiarity [37]; however, further consideration of kidney parameters, e.g., BUN and creatinine, along with subsequent histopathological examinations is important. Overall, the data from this study support the conclusion that the *L. martabanica* water leaf extract has non-toxic profiles.

The hematopoietic system is a sensitive indicator of drug toxicity in humans and animals, reflecting physiological and pathological conditions due to its role in nutrient distribution and substance transportation [40,41]. This study found no notable differences in hemoglobin, RBC count, or hematocrit between the untreated and *L. martabanica* water leaf extract-treated groups, suggesting no impact on blood cell production; however, the female rats showed significant changes in neutrophil, lymphocyte, and eosinophil levels in some groups. The male rats treated with the *L. martabanica* water leaf extract exhibited decreased WBC and lymphocyte counts, but all were within the acceptable clinical ranges [42].

Clinical serum biochemistry assessments were conducted to evaluate the renal and hepatic functions, focusing on markers such as BUN, creatinine, AST, ALT, ALP, total protein, albumin, and bilirubin. The kidneys, with their high blood perfusion, are susceptible to toxins through actively filtering and potentially accumulating harmful substances in renal tubules [43,44,45]. BUN and creatinine serve as sensitive markers of the renal condition, particularly in cases of renal or glomerular damage, where serum creatinine levels naturally rise [46,47]. Our study revealed no harmful impact on BUN or creatinine levels, indicating that the *L. martabanica* water leaf extract did not harm renal function in the female rats. While the male rats initially showed decreased BUN and creatinine levels at 2250 mg/kg, the levels returned to normal in the satellite group; however, BUN and creatinine in both the female and male rats remained within the standard ranges [48].

The liver plays a crucial role in digestion, detoxification, metabolism, and the removal of substances from the body [49]. Liver function tests are crucial diagnostic tools for detecting liver dysfunction. Bilirubin, a catabolic by-product of hemoglobin, is remarkably associated with hepatic diseases such as biliary cirrhosis and jaundice. High bilirubin levels are an indicator of the degree of hepatic dysfunction [44,50]. The synthetic or excretory capacity of the liver for total protein, albumin, globulin, and bilirubin is compromised when organ damage occurs [51]. AST, ALT, and alkaline phosphatase (ALP) are markers that are traditionally used to assess hepatic toxicity. In our investigation, after receiving the *L. martabanica* water leaf extract, albumin levels in female rats at 750 mg/kg, bilirubin levels at 250 and 750 mg/kg, and AST levels in the satellite-treatment group showed statistically significant increases but did not exceed the standard values [52]. Additionally, in male rats, no abnormalities were observed, suggesting no liver damage or disease, and indicating normal liver synthetic ability.

At the end of the study, according to both the WHO and OECD guidelines, histopathological assessments of the internal organs were conducted in the treated groups compared with those of the control group. All the internal organs, especially the heart, liver, kidneys, and lungs, revealed no abnormalities in cell morphology when compared between the control and the treated groups. The results of the present study consisting of body weight, organ weight, animal behaviors, hematological and serum biochemistry assessments, gross pathology, and histopathology assessments, when processed together, indicated that the *L. martabanica* water leaf extract did not cause chronic toxicity in rats.

## 5. Conclusions

The safety assessment of the *L. martabanica* water leaf extract, including the mutagenicity and acute and chronic toxicity assessments, showed no harmful effects. This study supports the clinical testing of the *L. martabanica* water leaf extract in humans and its integration into community healthcare systems. It aims to help integrate modern healthcare practices with highland communities’ traditional wisdom through a process of rigorous validation, using scientific processes to help expand accessible healthcare. Demonstrating the safety of traditional medications through these studies can provide crucial scientific evidence for informing public policy and enabling the commercial production and more widespread use of these natural materials by both highland and lowland populations.

## Figures and Tables

**Figure 1 toxics-12-00470-f001:**
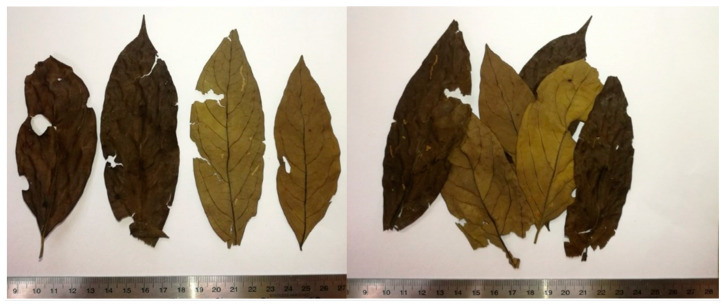
Macroscopic characteristics of *L. martabanica*.

**Figure 2 toxics-12-00470-f002:**
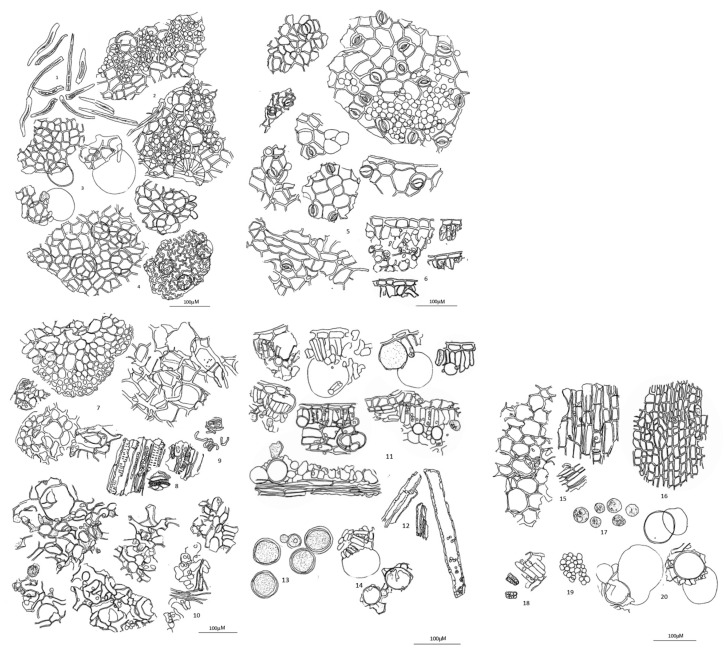
Diagnostic characteristics of the leaf of the *L. martabanica* powder: (1) Unicellular trichome filled with yellowish-brown pigment; (2) Upper epidermis showing the polygonal epidermis, lepidote, and unicellular trichomes with the palisade mesophyll and idioblast beneath; (3) Upper epidermis with idioblasts and what is clearly identifiable as excretory; (4) Upper epidermis with the idioblast scatter beneath; (5) Lower epidermis showing the tetracytic stomata, cicatrix, and mesophyll beneath; (6) Upper epidermis with the palisade mesophyll below in the sectional view; (7) Collenchyma in various views, some associated with the epidermis and spongy mesophyll; (8) Vascular strand; (9) Parts of the spiral vessel; (10) Spongy mesophyll; (11) Idioblast among the mesophyll in the sectional view; (12) Sclereids; (13) Idioblasts; (14) Idioblasts associated with the palisade mesophyll; (15) Parenchyma showing intercellular spaces and simple pits; (16) Epidermis showing the cicatrix; (17) Slightly opaque matter drops; (18) Palisade mesophyll; (19) Palisade mesophyll in surface view; (20) Idioblasts and what is clearly identifiable as excretory material.

**Figure 3 toxics-12-00470-f003:**
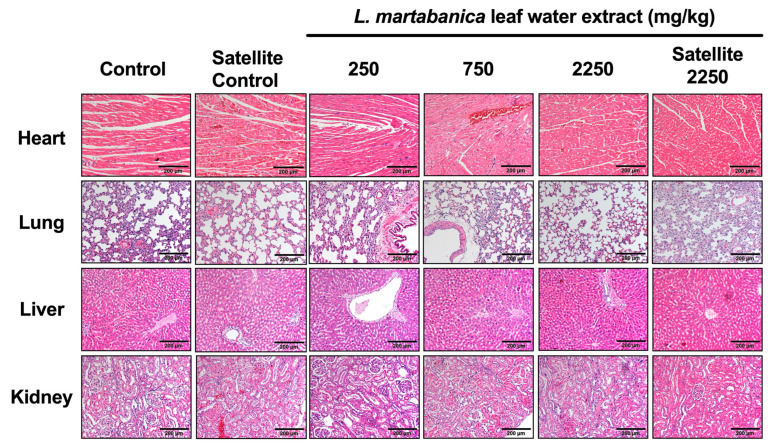
Histopathological assessment of the heart, lungs, liver, and kidneys from female rats using hematoxylin and eosin staining and imaging at 40× magnification.

**Figure 4 toxics-12-00470-f004:**
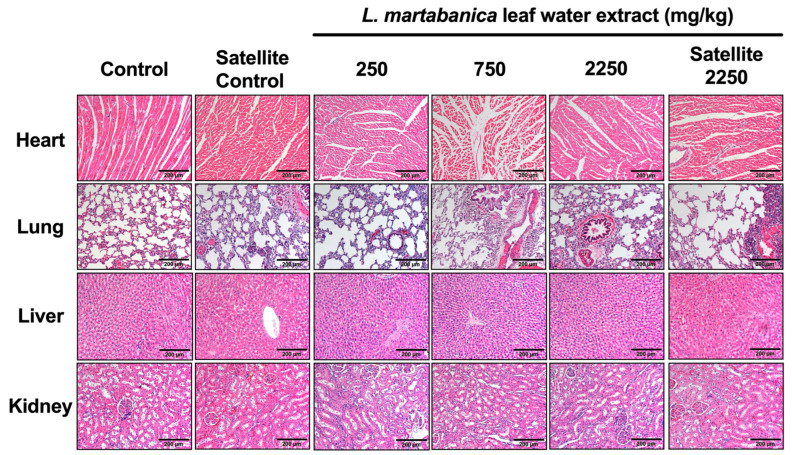
Histopathological assessment of the heart, lungs, liver, and kidneys from male rats using hematoxylin and eosin staining and imaging at 40× magnification.

**Table 1 toxics-12-00470-t001:** Pharmacognostic characters of the *L. martabanica* leaf.

Specification	Content (%) by Dried Weight
Loss on drying	Not more than 7
Total ash	Not more than 6
Acid-insoluble ash	Not more than 1
Ethanol-soluble extractive value	Not less than 9
Water–soluble extractive value	Not less than 13

**Table 2 toxics-12-00470-t002:** Mutagenicity assays for the *L. martabanica* water leaf extract using *Salmonella typhimurium* TA98 and TA100 strains with and without metabolic activation.

Sample	No. of Revertant Colony			
	*S. typhimurium* TA98		*S. typhimurium* TA100	
	+S9	−S9	+S9	−S9
DMSO	27 ± 1	24 ± 1	129 ± 3	125 ± 3
2-AA (0.5 µg/plate)	384 ± 8 #	-	478 ± 12 #	-
2-AF (0.1 µg/plate)	-	360 ± 11 #	-	445 ± 8 #
*L. martabanica* water leaf extract				
125 µg/plate	25 ± 2	21 ± 1	135 ± 3	128 ± 3
250 µg/plate	26 ± 2	21 ± 1	138 ± 2	126 ± 1
500 µg/plate	30 ± 1	25 ± 1	140 ± 4	129 ± 2
1000 µg/plate	30 ± 2	25 ± 1	138 ± 3	133 ± 4

The data are presented as mean ± S.E.M., *n* = 5. 2-AA; 2-aminoanthracene, 2-AF; 2-aminofluorene. # Significant difference from the DMSO group (*p* < 0.001).

**Table 3 toxics-12-00470-t003:** Results of the antimutagenicity testing of the *L. martabanica* water leaf extract on the *Salmonella typhimurium* TA98 and TA100 strains with and without metabolic activation.

Mutagen	Dose (µg/Plate)	Strain	S9	His^+^ Revertant Colonies/Plate Extract (µg/Plate) (%Inhibition)				
				0	125	250	500	1000
PhIP	0.1	TA98	+	307 ± 10 (0)	214 ± 8 (30) #	219 ± 6 (29) #	237 ± 7 (23) #	223 ± 6 (25) #
2-AF	0.1	TA98	−	459 ± 15 (0)	463 ± 5 (−1)	449 ± 8 (2)	467 ± 6 (−2)	452 ± 4 (1)
IQ	0.05	TA100	+	492 ± 24 (0)	419 ± 14 (15) *	390 ± 12 (21) #	357 ± 9 (27) #	317 ± 10 (36) #
2-AF	0.01	TA100	−	428 ± 15 (0)	344 ± 15 (20) **	353 ± 12 (18) **	377 ± 8 (12)	370 ± 9 (14) *

The data are presented as mean ± S.E.M. 2-AF; 2-aminofluorene, IQ; 2-amino-3-methyl-3H-imidazo[4,5-f] quinoline, PhIP; 2-amino-l-methyl-6-phenylimidazo[4,5-b] pyridine. *, **, and # Significant differences from the control (0 µg/Plate), *p* < 0.05, *p* < 0.01, and *p* < 0.001, respectively.

**Table 4 toxics-12-00470-t004:** The acute toxicity assessment: body weight of female rats.

Group	Body Weight (g)		
	Day 0	Day 7	Day 14
Control (2 mL/kg of distilled water)	208.00 ± 2.00	213.00 ± 2.55	221.00 ± 5.57
*L. martabanica* water leaf extract 5000 mg/kg	205.00 ± 3.16	214.00 ± 4.85	222.00 ± 10.32

The data are presented as mean ± S.E.M., *n* = 5 (female).

**Table 5 toxics-12-00470-t005:** The acute toxicity assessment: organ weight of female rats.

Organs	Female	
	Control (g)	*L. martabanica* Water Leaf Extract 5000 mg/kg (g)
Brain	1.97 ± 0.04	1.93 ± 0.05
Lung	1.18 ± 0.03	1.14 ± 0.05
Heart	0.72 ± 0.02	0.80 ± 0.04
Liver	7.24 ± 0.23	7.33 ± 0.41
Spleen	0.64 ± 0.05	0.64 ± 0.07
Adrenal gland	0.03 ± 0.00	0.04 ± 0.00
Kidney	0.91 ± 0.02	0.98 ± 0.04
Ovary	0.07 ± 0.00	0.07 ± 0.00
Uterus	0.46 ± 0.03	0.57 ± 0.05

The data are presented as mean ± S.E.M., *n* = 5 (female).

**Table 6 toxics-12-00470-t006:** The body weight of the female rats.

Day	Control	Satellite Control	*L. martabanica* Water Leaf Extract (mg/kg)			
			250	750	2250	Satellite 2250
1	207.00 ± 2.26	205.00 ± 2.24	208.00 ± 2.81	204.50 ± 2.17	207.00 ± 2.38	204.00 ± 1.87
30	257.00 ± 3.43	244.00 ± 2.92 *	254.50 ± 2.29	248.50 ± 3.66 *	257.00 ± 2.13	250.00 ± 4.47
90	321.50 ± 6.33	312.00 ± 3.39	311.50 ± 5.17	323.00 ± 6.11	320.00 ± 4.71	310.00 ± 9.80
120	331.50 ± 6.19	323.00 ± 5.83	324.00 ± 4.40	333.00 ± 5.93	330.00 ± 4.47	332.00 ± 9.30
180	352.00 ± 5.01	346.00 ± 3.32	344.50 ± 6.34	352.00 ± 4.30	347.00 ± 3.82	344.00 ± 6.40
240	365.00 ± 4.28	352.00 ± 2.55	354.00 ± 6.14	359.00 ± 4.52	353.50 ± 4.60	347.00 ± 5.15
270	379.50 ± 8.55	376.00 ± 5.10	366.00 ± 7.92	380.00 ± 6.32	376.00 ± 6.78	364.00 ± 12.19
298	-	376.00 ± 9.67	-	-	-	350.00 ± 16.12

The data are presented as mean ± S.E.M., *n* = 10, *n* = 5 (satellite groups). * Significant difference from the control (*p* < 0.05).

**Table 7 toxics-12-00470-t007:** The body weight of the male rats.

Day	Control	Satellite Control	*L. martabanica* Water Leaf Extract (mg/kg)			
			250	750	2250	Satellite 2250
1	231.50 ± 2.99	226.00 ± 3.67	234.00 ± 2.21	236.00 ± 2.21	233.50 ± 2.99	227.00 ± 3.74
30	312.00 ± 2.91	322.00 ± 4.90	319.00 ± 7.22	327.00 ± 3.27 *	317.50 ± 3.44	337.00 ± 8.60 *
90	429.50 ± 7.09	438.00 ± 13.66	436.00 ± 14.47	441.50 ± 8.37	435.00 ± 6.32	459.00 ± 16.76
120	443.00 ± 6.20	454.00 ± 13.73	450.00 ± 14.02	456.00 ± 6.45	453.50 ± 6.83	475.00 ± 8.66 *
180	467.50 ± 7.08	475.00 ± 11.18	470.50 ± 10.99	482.00 ± 5.28	478.00 ± 6.15	498.00 ± 4.90 *
240	480.50 ± 5.89	479.00 ± 9.67	479.50 ± 9.79	490.00 ± 4.77	486.00 ± 4.76	510.00 ± 2.74 *
270	513.50 ± 11.28	516.00 ± 24.82	539.50 ± 14.09	526.50 ± 9.04	545.00 ± 10.75	553.00 ± 22.67
298	-	510.00 ± 18.64	-	-	-	524.00 ± 14.00

The data are presented as mean ± S.E.M., *n* = 10, *n* = 5 (satellite groups). * Significant difference from the control (*p* < 0.05).

**Table 8 toxics-12-00470-t008:** The relative organ weight (g% body weight) of the female rats.

Organ	Control	Satellite Control	*L. martabanica* Water Leaf Extract (mg/kg)			
			250	750	2250	Satellite 2250
Brain	0.69 ± 0.02	0.59 ± 0.01	0.71 ± 0.01	0.63 ± 0.01	0.65 ± 0.06	0.62 ± 0.02
Lung	0.63 ± 0.03	0.59 ± 0.05	0.66 ± 0.08	0.68 ± 0.05	0.65 ± 0.03	0.58 ± 0.03
Heart	0.36 ± 0.01	0.36 ± 0.01	0.38 ± 0.01	0.38 ± 0.01	0.38 ± 0.01	0.36 ± 0.01
Liver	4.18 ± 0.10	4.49 ± 0.20	4.13 ± 0.08	3.98 ± 0.11	4.25 ± 0.14	4.48 ± 0.19
Spleen	0.18 ± 0.01	0.20 ± 0.01	0.19 ± 0.01	0.18 ± 0.00	0.20 ± 0.01	0.18 ± 0.01
Adrenal gland	0.01 ± 0.00	0.01 ± 0.00	0.01 ± 0.00	0.01 ± 0.00	0.01 ± 0.00	0.01 ± 0.00
Kidney	0.48 ± 0.02	0.42 ± 0.01	0.46 ± 0.01	0.39 ± 0.01 *	0.41 ± 0.01	0.43 ± 0.02
Ovary	0.02 ± 0.00	0.02 ± 0.00	0.02 ± 0.00	0.02 ± 0.00	0.02 ± 0.00	0.02 ± 0.00
Uterus	0.32 ± 0.02	0.29 ± 0.02	0.28 ± 0.02	0.27 ± 0.02	0.33 ± 0.07	0.29 ± 0.03

The data are presented as mean ± S.E.M., *n* = 10, *n* = 5 (satellite groups). * Significant difference from the control (*p* < 0.05).

**Table 9 toxics-12-00470-t009:** The relative organ weight (g% body weight) of the male rats.

Organ	Control	Satellite Control	*L. martabanica* Water Leaf Extract (mg/kg)			
			250	750	2250	Satellite 2250
Brain	0.50 ± 0.02	0.48 ± 0.02	0.51 ± 0.02	0.50 ± 0.02	0.47 ± 0.01	0.42 ± 0.03
Lung	0.55 ± 0.03	0.46 ± 0.03	0.55 ± 0.02	0.53 ± 0.02	0.57 ± 0.04	0.43 ± 0.04
Heart	0.32 ± 0.01	0.35 ± 0.04	0.31 ± 0.02	0.32 ± 0.01	0.31 ± 0.01	0.30 ± 0.01
Liver	3.58 ± 0.12	3.04 ± 0.17	3.58 ± 0.16	3.21 ± 0.31	3.38 ± 0.09	2.95 ± 0.13
Spleen	0.19 ± 0.12	0.15 ± 0.01	0.16 ± 0.01	0.18 ± 0.01	0.17 ± 0.01	0.14 ± 0.01
Adrenal glands	0.01 ± 0.00	0.01 ± 0.00	0.01 ± 0.00	0.01 ± 0.00	0.01 ± 0.00	0.01 ± 0.00
Kidney	0.40 ± 0.02	0.36 ± 0.03	0.39 ± 0.02	0.39 ± 0.01	0.39± 0.01	0.31 ± 0.01 *
Testis	0.39 ± 0.01	0.37 ± 0.02	0.36 ± 0.01	0.38 ± 0.01	0.35 ± 0.01	0.36 ± 0.02
Epididymis	0.19 ± 0.01	0.18 ± 0.01	0.23 ± 0.02	0.19 ± 0.01	0.19 ± 0.01	0.16 ± 0.01

The data are presented as mean ± S.E.M., *n* = 10, *n* = 5 (satellite groups). * Significant difference from the control (*p* < 0.05).

**Table 10 toxics-12-00470-t010:** The hematological assessment of the female rats.

Blood Parameters	Control	Satellite Control	*L. martabanica* Water Leaf Extract (mg/kg)			
			250	750	2250	Satellite 2250
Red blood cell (×10^6^/μL)	7.55 ± 0.07	7.37 ± 0.11	7.30 ± 0.08	7.35 ± 0.16	7.26 ± 0.14	7.45 ± 0.12
Hemoglobin (g/dL/L)	14.08 ± 0.14	14.24 ± 0.12	13.76 ± 0.19	14.05 ± 0.22	13.91 ± 0.25	14.26 ± 0.10
Hematocrit (%)	41.46 ± 0.45	41.36 ± 0.33	40.41 ± 0.61	41.36 ± 0.76	40.72 ± 0.77	42.00 ± 0.30
Mean corpuscular volume (fL)	54.88 ± 0.42	56.16 ± 0.99	55.38 ± 0.82	56.35 ± 0.47	56.14 ± 0.77	56.40 ± 1.20
Mean corpuscular hemoglobin (pg)	18.64 ± 0.19	19.36 ± 0.31	18.87 ± 0.26	19.14 ± 0.13	19.18 ± 0.25	19.14 ± 0.40
Mean corpuscular hemoglobin concentration (g/dL)	33.98 ± 0.19	34.50 ± 0.16	34.05 ± 0.11	33.99 ± 0.19	34.15 ± 0.09	33.96 ± 0.10
Platelet (×10^5^/μL)	8.32 ± 0.30	7.84 ± 0.21	8.24 ± 0.24	8.05 ± 0.17	8.15 ± 0.26	7.73 ± 0.21
White blood cells (×10^3^/μL)	3.99 ± 0.30	3.24 ± 0.20	4.04 ± 0.37	3.21 ± 0.25	4.11 ± 0.27	3.83 ± 0.15
Neutrophil (×10^3^/μL)	1.34 ± 0.13	0.99 ± 0.07	1.69 ± 0.19	1.17 ± 0.13	1.80 ± 0.18 *	1.20 ± 0.08
Lymphocyte (×10^3^/μL)	2.06 ± 0.19	1.95 ± 0.20	1.59 ± 0.16 *	1.43 ± 0.15 *	1.76 ± 0.14	2.35 ± 0.14
Monocyte (×10^3^/μL)	0.43 ± 0.08	0.20 ± 0.01	0.61 ± 0.14	0.49 ± 0.08	0.40 ± 0.09	0.32 ± 0.05
Eosinophil (×10^3^/μL)	0.17 ± 0.02	0.16 ± 0.01	0.17 ± 0.02	0.09 ± 0.01 *	0.15 ± 0.02	0.14 ± 0.01
Basophil (×10^3^/μL)	0.00 ± 0.00	0.00 ± 0.00	0.00 ± 0.00	0.00 ± 0.00	0.00 ± 0.00	0.00 ± 0.00

The data are presented as mean ± S.E.M., *n* = 10, *n* = 5 (satellite groups). * Significant difference from the control (*p* < 0.05).

**Table 11 toxics-12-00470-t011:** The hematological assessment of the male rats.

Blood Parameters	Control	Satellite Control	*L. martabanica* Water Leaf Extract (mg/kg)			
			250	750	2250	Satellite 2250
Red blood cell (×10^6^/μL)	8.24 ± 0.10	8.64 ± 0.09	8.16 ± 0.18	8.33 ± 0.09	8.38 ± 0.16	8.60 ± 0.09
Hemoglobin (g/dL)	15.16 ± 0.12	15.42 ± 0.23	14.67 ± 0.28	15.10 ± 0.13	14.84 ± 0.18	15.56 ± 0.24
Hematocrit (%)	44.54 ± 0.46	45.66 ± 0.56	43.45 ± 0.86	44.91 ± 0.57	44.08 ± 0.55	46.06 ± 0.84
Mean corpuscular volume (fL)	54.07 ± 0.64	52.82 ± 0.27	53.34 ± 0.60	53.95 ± 0.50	52.70 ± 0.62	53.56 ± 0.67
Mean corpuscular hemoglobin (pg)	18.42 ± 0.20	17.84 ± 0.10	18.00 ± 0.17	18.15 ± 0.16	17.95 ± 0.19	18.10 ± 0.20
Mean corpuscular hemoglobin concentration (g/dL)	34.06 ± 0.27	33.78 ± 0.18	33.77 ± 0.11	33.64 ± 0.18	33.65 ± 0.13	33.80 ± 0.15
Platelet (×10^5^/μL)	7.87 ± 0.24	7.99 ± 0.23	7.62 ± 0.23	7.85 ± 0.26	8.09 ± 0.35	8.17 ± 0.21
White blood cells (×10^3^/μL)	6.06 ± 0.22	6.25 ± 0.25	5.19 ± 0.38	4.88 ± 0.18 *	5.39 ± 0.36	5.61 ± 0.54
Neutrophil (×10^3^/μL)	1.27 ± 0.15	1.73 ± 0.26	1.41 ± 0.19	1.19 ± 0.08	1.70 ± 0.19	1.49 ± 0.14
Lymphocyte (×10^3^/μL)	4.20 ± 0.23	4.08 ± 0.10	2.99 ± 0.24 *	2.83 ± 0.18 *	2.88 ± 0.17 *	3.30 ± 0.34 *
Monocyte (×10^3^/μL)	0.42 ± 0.09	0.29 ± 0.02	0.60 ± 0.14	0.68 ± 0.07	0.61 ± 0.13	0.61 ± 0.21
Eosinophil (×10^3^/μL)	0.17 ± 0.01	0.14 ± 0.02	0.19 ± 0.03	0.18 ± 0.02	0.20 ± 0.02	0.20 ± 0.02
Basophil (×10^3^/μL)	0.00 ± 0.00	0.00 ± 0.00	0.00 ± 0.00	0.00 ± 0.00	0.00 ± 0.00	0.00 ± 0.00

The data are presented as mean ± S.E.M., *n* = 10, *n* = 5 (satellite groups). * Significant difference from the control (*p* < 0.05).

**Table 12 toxics-12-00470-t012:** The serum biochemistry assessment of the female rats.

Blood Parameters	Control	Satellite Control	*L. martabanica* Water Leaf Extract (mg/kg)			
			250	750	2250	Satellite 2250
BUN (mg/dL)	14.76 ± 0.77	15.12 ± 2.26	15.70 ± 1.02	14.83 ± 0.35	16.56 ± 0.72	15.22 ± 0.43
Creatinine (mg/dL)	0.80 ± 0.02	0.79 ± 0.01	0.76 ± 0.03	0.77 ± 0.02	0.78 ± 0.02	0.73 ± 0.01
Total protein (g/dL)	7.23 ± 0.12	7.24 ± 0.10	7.30 ± 0.08	7.44 ± 0.11	7.12 ± 0.08	7.16 ± 0.11
Albumin (g/dL)	3.98 ± 0.06	3.90 ± 0.03	3.95 ± 0.06	4.15 ± 0.05 *	3.89 ± 0.05	4.04 ± 0.02
Total bilirubin (mg/dL)	0.17 ± 0.01	0.15 ± 0.02	0.20 ± 0.01 *	0.21 ± 0.01 *	0.18 ± 0.01	0.17 ± 0.00
Direct bilirubin (mg/dL)	0.05 ± 0.01	0.05 ± 0.02	0.07 ± 0.00	0.06 ± 0.00	0.06 ± 0.00	0.07 ± 0.00
AST (U/L)	60.60 ± 4.95	65.00 ± 9.17	74.30 ± 6.95	70.60 ± 4.43	72.10 ± 6.59	82.60 ± 10.47 *
ALT (U/L)	28.10 ± 3.45	35.60 ± 4.01	35.80 ± 4.43	34.60 ± 3.11	33.80 ± 3.58	39.20 ± 7.37
ALP (U/L)	22.10 ± 1.74	22.00 ± 5.61	17.20 ± 0.93	17.80 ± 0.95	21.00 ± 1.91	22.80 ± 0.58

The data are presented as mean ± S.E.M., *n* = 10, *n* = 5 (satellite groups). * Significant difference from the control (*p* < 0.05). BUN, blood urea nitrogen; AST, aspartate aminotransferase; ALT, alanine aminotransferase; ALP, alkaline phosphatase.

**Table 13 toxics-12-00470-t013:** The serum biochemistry assessment of the male rats.

Blood Parameters	Control	Satellite Control	*L. martabanica* Water Leaf Extract (mg/kg)			
			250	750	2250	Satellite 2250
BUN (mg/dL)	15.22 ± 0.79	14.36 ± 0.32	14.82 ± 1.02	13.62 ± 0.61	13.08 ± 0.38 *	15.40 ± 0.64
Creatinine (mg/dL)	0.70 ± 0.02	0.73 ± 0.01	0.66 ± 0.02	0.64 ± 0.01 *	0.63 ± 0.01 *	0.71 ± 0.02
Total protein (g/dL)	5.81 ± 0.08	5.92 ± 0.10	5.95 ± 0.09	5.84 ± 0.07	5.80 ± 0.06	5.76 ± 0.10
Albumin (g/dL)	2.98 ± 0.03	2.96 ± 0.04	3.04 ± 0.03	3.00 ± 0.04	2.96 ± 0.02	2.94 ± 0.05
Total bilirubin (mg/dL)	0.16 ± 0.01	0.20 ± 0.06	0.16 ± 0.00	0.16 ± 0.01	0.16 ± 0.01	0.15 ± 0.02
Direct bilirubin (mg/dL)	0.06 ± 0.00	0.06 ± 0.01	0.06 ± 0.00	0.07 ± 0.00	0.07 ± 0.00	0.07 ± 0.00
AST (U/L)	72.90 ± 2.45	86.20 ± 5.75	84.70 ± 8.14	87.50 ± 8.71	78.30 ± 2.79	81.80 ± 3.38
ALT (U/L)	28.40 ± 1.97	38.80 ± 3.34	36.20 ± 5.36	36.60 ± 5.54	26.80 ± 1.02	34.60 ± 3.09
ALP (U/L)	64.30 ± 4.03	71.20 ± 3.26	65.60 ± 3.67	69.20 ± 7.20	59.40 ± 3.87	64.20 ± 3.51

The data are presented as mean ± S.E.M., *n* = 10, *n* = 5 (satellite groups). * Significant difference from the control (*p* < 0.05). BUN, blood urea nitrogen; AST, aspartate aminotransferase; ALT, alanine aminotransferase; ALP, alkaline phosphatase.

## Data Availability

Data are available upon request.

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
