# Peer review of "The Safety Assessment of Mutagenicity, Acute and Chronic Toxicity of the Litsea martabanica (Kurz) Hook.f. Water Leaf Extract"

_toxics, 2024, doi:10.3390/toxics12070470_

Round 1

Reviewer 1 Report

Comments and Suggestions for Authors

Dear Authors 

this is an interesting article regarding the safety of Litsea martabanica leaf extracts with animal tests on Sprague-Dawley rats.

I kindly ask the authors to include the phytochemical composition of the extract, to detail the correlations between chemical composition and bioactivity and with the existing literature.

Please ensure that the botanical names throughout the manuscript are uniformly italicized, as they sometimes appear without this formatting.

Warmest regards

Comments on the Quality of English Language

The overall English is good. I recommend a thorough review of the final draft to prevent typographical errors.

Author Response

  1. I kindly ask the authors to include the phytochemical composition of the extract, to detail the correlations between chemical composition and bioactivity and with the existing literature.

Response: We have already added the details of chemical compositions and bioactivities in the discussion part (Lines 493-499).

Reference:

Jimenez-Lopez, C.; Pereira, A.G.; Lourenço-Lopes, C.; Garcia-Oliveira, P.; Cassani, L.; Fraga-Corral, M.; Prieto, M.A.; Simal-Gandara, J. Main bioactive phenolic compounds in marine algae and their mechanisms of action supporting potential health benefits. Food Chemistry 2021, 341(Pt 2), 128262, doi: 10.1016/j.foodchem.2020.128262. (ref. 25)

Dias, M.C.; Pinto, D.C.G. A.; Silva, A.M.S. Plant Flavonoids: Chemical Characteristics and Biological Activity. Molecules 2021, 26(17), 5377, doi: 10.3390/molecules26175377 (ref. 26)

Zhao, W.Y.; Yi, J.; Chang, Y.B.; Sun, C.P.; Ma, X.C. Recent studies on terpenoids in Aspergillus fungi: Chemical diversity, biosynthesis, and bioactivity. Phytochemistry 2022, 193, 113011, doi: 10.1016/j.phytochem.2021.113011 (ref. 27)

Nguyen, L.T.; Fărcaş, A.; Socaci, S.; Tofana, M.; Diaconeasa, Z.; Pop, O.L.; Salanță, L.C. An Overview of Saponins -A Bioactive Group. Bulletin UASVM Food Science and Technology 2020, 77(1), doi: 10.15835/buasvmcn-fst:2019.0036 (ref. 28)

  1. Please ensure that the botanical names throughout the manuscript are uniformly italicized, as they sometimes appear without this formatting.

Response: Thank you for your pointing out. We have already italicized and carefully checked all the botanical names throughout the manuscript.

Reviewer 2 Report

Comments and Suggestions for Authors

In this manuscript, the safety of Litsea martabanica (Kurz) Hook.f. leaf water exracts were checked via mutagenicity tests, acute and chronic toxicity tests. In the Ames test, no mutagenic activation was observed. In vivo acute and chronic toxicity tests in Sprague-Dawley rats revealed no adverse effects on rats after 14 days of 5000 mg/kg single dose oral administration, and over 270 days of daily oral administration. The current manuscript could be accepted upon revision.

1. The acute toxicity evaluation, it is suggested to provide rationale for that only female rats were recruited.

2. The authors stated that acute toxicity indicators were monitored daily for 14 days for the acute toxicity evaluation. However, these data were not presented in the manuscript. The behavioral patterns, and clinical indicators for the chronic toxicity evaluation are not presented either.

3. The results should be presented as mean ± S.D. since each value should represent one rat.

4. As shown in Table 11, the basophil is presented as 0.00±0.00, which is not commonly seen. Although basophils are rare population, the author could change the unit for the counting, such as x10/uL. There are some other places that the data are presented as the mean ±0.00, please revise accordingly.

5. According to the H&E staining results of the lung, it seems some group may not be normal, with condensed alveoli. Enlarged images should be presented, and may consult with clinicians for the diagnosis.

Minor points:

1. The authors may want to check althrough the manuscript for typos, such as the mean ±S.E.M. were typed as the mean ±S.M.E. in many places. 

Comments on the Quality of English Language

Need to be improved.

Author Response

  1. The acute toxicity evaluation, it is suggested to provide rationale for that only female rats were recruited.

Respones: Our test design is based on both guidelines (WHO and OECD). The WHO guideline indicates on page 28 in the section on acute toxicity, "In at least one of the species, males and females should be used." with the following limitation: "In the case of rodents, each group should consist of at least five animals per sex." Whereas the OECD 420 guideline states in paragraph 1 that "testing in one sex (usually females) is now considered sufficient." In our experiment, we combined the WHO and OECD guidelines and used only female subjects. It is well known that females are more sensitive to any toxic substance than males. Typically, females are tested first because if a substance is not toxic to females, it is generally not toxic to males. Furthermore, the Animal Ethics Committee agreed that this section should be carried out in accordance with firstly by the OECD guideline and with the use of single-gender female experimental animals. The rationale was added in the discussion part (lines 532-537).

  1. The authors stated that acute toxicity indicators were monitored daily for 14 days for the acute toxicity evaluation. However, these data were not presented in the manuscript. The behavioral patterns, and clinical indicators for the chronic toxicity evaluation are not presented either. 

Respones:

Following the guidelines outlined by WHO 2000 (acute toxicity on page 28 and chronic toxicity on page 30) and OECD (acute toxicity 420 on page 6 and chronic toxicity 452 on page 14), the trials examined behavioral patterns and clinical indicators. Researchers adhered to the principles of Hippocratic screening. Acute toxicity symptoms were monitored for 14 days, while chronic toxicity indicators were observed for 270 days, extending to 298 days in the satellite group. We have added information as mentioned in the materials and methods (Lines 218-228), results (Lines 335-339), and discussion (Lines 527-528 and 546). In addition, Tables S1 and S2 (Table S1: Hippocratic screening for acute toxicity of L. martabanica leaf extract in female rats and Table S2: Hippocratic screening for chronic toxicity of L. martabanica leaf extract in female and male rats) have been added as supplementary data.

  1. The results should be presented as mean ± S.D. since each value should represent one rat. 

Respones: Standard deviation (S.D.) is used to indicate the distribution of a population. The standard error of the mean (S.E.M.) is used to determine the stability of the mean obtained from a random sample. S.E.M. indicates the quality of sampling, specifically how reliable the average obtained from a random sample of the population is. Therefore, researchers have chosen to use S.E.M. to present the statistics in this study.

  1. As shown in Table 11, the basophil is presented as 0.00 ± 0.00, which is not commonly seen. Although basophils are rare population, the author could change the unit for the counting, such as x10/uL. There are some other places that the data are presented as the mean ± 0.00, please revise accordingly. 

Respones: For hematology analysis, whole blood was collected in ethylenediaminetetraacetic acid (EDTA) containing tubes and measured using an automated hematology analyzer (BC-5300 Vet, Mindray, Shenzhen, China) provided by the Small Animal Hospital at Chiang Mai University. The results were reported by the veterinary technologist and clinical veterinarian. Normally, for this species of SD rat, we might rely on the standard values ​​of the National Laboratory Animal Center of Mahidol University. However, when compared with the reference of Delwatta, Shehani L., et al., this value is still considered normal. Hematological parameters are normally reported not in the form x10/mL, but in the form of percentage (%) or x102/mL - x106/mL

Reference: Delwatta, Shehani L., et al. "Reference values for selected hematological, biochemical and physiological parameters of Sprague‐Dawley rats at the Animal House, Faculty of Medicine, University of Colombo, Sri Lanka." Animal models and experimental medicine 1.4 (2018): 250-254.

  1. According to the H&E staining results of the lung, it seems some group may not be normal, with condensed alveoli. Enlarged images should be presented, and may consult with clinicians for the diagnosis. 

 Respones: Results of various experiments consisting of hematological and biochemical parameters, gross pathology, and histopathology were interpreted and compared by medical pathologists and veterinary pathologists. However, the difference in the images was thought by pathologists to be the result of different dimensions and sections in the tissue. Normally, if animals are exposed to a test substance, the toxicity of that substance to the lungs might be assessed along with other characteristics such as necrosis, inflammation, constricted bronchi and vessels, congestion of vessels, hyperplasic distended and widely open arteries, thickening of alveoli, constricted and appearance of papillary projections into lumen bronchioles. However, in this study, there were no such abnormalities in the lung tissue. In addition, we have enlarged the images to make it clearer.

Minor points:

  1. The authors may want to check althrough the manuscript for typos, such as the mean ±S.E.M. were typed as the mean ±S.M.E. in many places. 

Respones: We have carefully checked and edited all S.M.E. to S.E.M.

Reviewer 3 Report

Comments and Suggestions for Authors

This manuscript describes the study of acute and chronic toxicity of water  extracts from the leaves of Litsea martabanica (Kurz) Hook.f., a Lauraceae traditionally used as anti-insecticidal and hepatoprotective.

The title could be improved, it is repetitive.

Keywords such as Animal and Access to healthcare should be removed.
At the introduction, the first paragraph has no references. Overall, the introduction is well-formulated, brings a contextualization and the main objectives of the study.

References are out of format and need to be updated, with only 10% of self-citations.

Materials and methods allow the clear observation of the procedures,with appropriate references, but not at the pharmacognostic assays.

Italics should be observed every time the plant name is cited, such as at line 226.

The pharmacognostic assays are very interesting but they did not appear at the abstract, nor the introduction and could be better presented at the methodology.

Tables 2, 3, 5 and 6 need appropriate statistical significance. Table 4 could be excluded. Statistics at tables 7 and 8 could be improved.

The first two paragraphs of discussion are not about the results, but the study, and could eliminated of moved to introduction.

At lines 460-1authors insert an information about phytochemical analysis that are not present at the study, and with no references. This paragraph and the next one are also out of context and are not linked with the results obtained

The last paragraph of the discussion, 551-563 is a conclusion. Should be excluded or merged with the conclusion.

Comments on the Quality of English Language

None.

Author Response

  1. The title could be improved, it is repetitive.

Response: We have changed the title of our article to be “Mutagenicity, Acute and Chronic Toxicity Evaluation of the Water Extract from the Leaf of Litsea martabanica (Kurz) Hook.f.”

  1. Keywords such as Animal and Access to healthcare should be removed.

Response: We have considered that “Environment and human health” should be added while “Animal” and “Access to healthcare” should be removed from our manuscript.

  1. At the introduction, the first paragraph has no references. Overall, the introduction is well-formulated, brings a contextualization and the main objectives of the study.

Response: We greatly appreciate your suggestion and have considered incorporating the appropriate quotations or references into the initial introduction section.

Reference:

Kong, D.G.; Zhao, Y.; Li, G.H.; Chen, B.J.; Wang, X.N.; Zhou, H.L.; Lou, H.X.; Ren, D.M.; Shen T. The genus Litsea in traditional Chinese medicine: An ethnomedical, phytochemical and pharmacological review. Journal of Ethnopharmacology 2015, 164, 256–264, doi: 10.1016/j.jep.2015.02.020 (ref. 1)

Thielmann, J.; Muranyi, P. Review on the chemical composition of Litsea cubeba essential oils and the bioactivity of its major constituents citral and limonene. Journal of Essential Oil Research 2019, 31 (5), 361–378, doi: 10.1080/10412905.2019.1611671 (ref. 2)

Ho, C.L.; Jie-Pinge, O.; Liu, Y.C.; Hung, C.P.; Tsai, M.C.; Liao, P.C.; Wang, E.I.; Chen, Y.L.; Su, Y.C. Compositions and in vitro anticancer activities of the leaf and fruit oils of Litsea cubeba from Taiwan. Natural Product Communications 2010, 5(4), 617-620. (ref. 3)

Shen, G.D.; Zhang, Y.Y.; Yang, N.Q.; Yang, T.; Wang, T.; Lu, S.C.; Wang, J.Y.; Wang, Y.S.; Yang, J.H. N-alkylamides from Litsea cubeba (Lour.) Pers. With potential anti-inflammatory activity. Natural Product Research 2024, 38(10), 1727–1738, doi: 10.1080/14786419.2023.2222216 (ref. 4)

Kamle, M.; Mahato, D.K.; Lee, K.E.; Bajpai, V.K.; Gajurel, P.R.; Gu, K.S.; Kumar, P. Ethnopharmacological properties and medicinal uses of Litsea cubeba. Plants 2019, 8, 150. (ref. 5)

  1. References are out of format and need to be updated, with only 10% of self-citations.

Response: We updated several of the references in our manuscript.

  1. Materials and methods allow the clear observation of the procedures, with appropriate references, but not at the pharmacognostic assays.

Response: Thank you for your pointing out. We have added methods to the pharmacognostic assays in sections 2.3.1 - 2.3.3 as appropriate (Lines 128-163).

  1. Italics should be observed every time the plant name is cited, such as at line 226.

Response: We have already italicized the botanical names throughout the manuscript.

  1. The pharmacognostic assays are very interesting but they did not appear at the abstract, nor in the introduction and could be better presented at the methodology.

Response: Thank you for your recommendation. We have included the description of pharmacognostic assays in the abstract (Lines 26-28), introduction (Lines 85-88), and methods (Lines 128-163).

  1. Tables 2, 3, 5 and 6 need appropriate statistical significance. Table 4 could be excluded. Statistics at Tables 7 and 8 could be improved.

Response:

  1. Following statistical analysis, it was discovered that many of the data points in Tables 2 and 3 were statistically significant. As a result, we have appropriately revised the data and the description in the results section (Lines 310-312 (Table 2), and 322-328 (Table 3)).
  2. For Table 4, we have already removed, and these tables have been modified and added as supplementary data Tables S1 and S2 (Table S1: Hippocratic screening for acute toxicity of martabanica leaf extract in female rats and Table S2: Hippocratic screening for chronic toxicity of L. martabanica leaf extract in female and male rats, respectively).
  3. For Tables 5 and 6, there were no significant changes in any parameters. Thus, this part would not be edited.
  4. For Tables 7 and 8, we have carefully checked, statistically investigated, and corrected the information.
  5. The first two paragraphs of discussion are not about the results, but the study, and could eliminated of moved to the introduction.

Response: We have already deleted the first paragraph and some parts of the second paragraph. Then, we have summarized and rewrote it (Lines 475-477).

  1. At lines 460-1authors insert an information about phytochemical analysis that are not present at the study, and with no references. This paragraph and the next one are also out of context and are not linked with the results obtained

Response: We have included information about extraction in this study to the methodology in section 2.4. (Lines 165-166) to keep the content relevant to the discussion (Lines 475-477).

  1. The last paragraph of the discussion, 551-563 is a conclusion. Should be excluded or merged with the conclusion.

Response: We have removed some sentences and rewrote the last paragraph of the discussion part (Lines 591-598).

Round 2

Reviewer 1 Report

Comments and Suggestions for Authors

Dear authors, Thank you for addressing the previous suggestions. However, the phytochemical composition is still missing. Kindly include a table with the main chemical constituents present in the extract. Reference number 7 does not report this information. If you are unable to analyze your extract, please provide the data present in the literature. 

Thank you.nt in the literature. 

Thank you.

Author Response

Reviewer 1

Comments and Suggestions for Authors

Dear authors, Thank you for addressing the previous suggestions. However, the phytochemical composition is still missing. Kindly include a table with the main chemical constituents present in the extract. Reference number 7 does not report this information. If you are unable to analyze your extract, please provide the data present in the literature. 

Response:

Phytochemical screening identifies different classes of phytoconstituents present in various parts of a plant. The confirmatory qualitative phytochemical screening of plant extracts was performed to identify the main classes of compounds (tannins, saponins, flavonoids, alkaloids, phenols, glycosides, steroids, and terpenoids) present in the extracts following standard protocols. Phytochemical studies typically involve chemical or color reactions. These methods and characteristics indicate the quality of standard extracts used for testing in the research.

Although phytochemical screening cannot elucidate the chemical structures of isolated constituents, it is the basis for further study to isolate the bioactive or chemical marker via bioassay-guided isolation.

Therefore, we have modified the information in the results section (Section 3.1.3. Physical and Chemical Identification, lines 303 - 305) to provide more reference about phytochemical screening, which showed the presence of phenolics, flavonoids, saponins, and terpenoids.

Ref: Lawal AM, Lawan MM, Apampa SA. Phytochemical analysis and thin layer chromatography profiling of crude extracts from Guiera senegalensis (Leaves). J Biotechnol Biomed Sci. 2019;3(3):7–12. 

Reviewer 3 Report

Comments and Suggestions for Authors

Authors presented a new reformulated version, observing several issues pointed out by reviewers. Indeed, at the new texts are some adjustments needed, such as the repetitive text at lines 492-496.

Also, references 28 and 47 are still out of format,

Comments on the Quality of English Language

None.

Author Response

Reviewer 3

Comments and Suggestions for Authors

Authors presented a new reformulated version, observing several issues pointed out by reviewers. Indeed, at the new texts are some adjustments needed, such as the repetitive text at lines 492-496.

Also, references 28 and 47 are still out of format,

            Response:

In the introduction, we discuss plants in the Lauraceae family, using L. cubeba as an example. L. cubeba has antioxidant, anti-cancer, anti-inflammatory, insecticidal, and hepatoprotective properties. This leads us to believe that L. martabanica, which grows in high-altitude areas and also belongs to Lauraceae family, should exhibit similar effects, especially insecticidal and hepatoprotective properties. These effects align with traditional knowledge from high-altitude regions, inspiring our interest and confidence in studying this plant. In lines 492-496 of the discussion, we address a group of constituents that can have various biological effects, suggesting that L. martabanica also contains these extracts. Therefore, it is possible that L. martabanica works as described in the literature review.

In addition, all references have been carefully reviewed and corrected. Reference 28 has been changed to 29, and reference 47 has been changed to 48, in accordance with the MDPI format.
